# Dental Appliances for the Treatment of Obstructive Sleep Apnea in Children: A Systematic Review and Meta-Analysis

**DOI:** 10.3390/medicina59081447

**Published:** 2023-08-10

**Authors:** Daniel Marciuc, Stefan Morarasu, Bianca Codrina Morarasu, Emilia Adriana Marciuc, Bogdan Ionut Dobrovat, Veronica Pintiliciuc-Serban, Roxana Mihaela Popescu, Florinel Cosmin Bida, Valentin Munteanu, Danisia Haba

**Affiliations:** 1Surgery Department, Faculty of Dental Medicine, “Grigore T. Popa” University of Medicine and Pharmacy, 700115 Iasi, Romania; d.marciuc@yahoo.com (D.M.); veronica.serban@yahoo.com (V.P.-S.); drpopescuroxana@yahoo.com (R.M.P.); 22nd Department of Surgical Oncology, Regional Institute of Oncology, Faculty of Medicine, “Grigore T. Popa” University of Medicine and Pharmacy, 700115 Iasi, Romania; morarasu.stefan@gmail.com; 3Department of Internal Medicine and Toxicology, “Saint Spiridon” University Regional Emergency Hospital, Faculty of Medicine, “Grigore T. Popa” University of Medicine and Pharmacy, 700115 Iasi, Romania; 4Department of Radiology, Emergency Hospital “Prof. Dr. Nicolae Oblu”, Faculty of Dental Medicine, “Grigore T. Popa” University of Medicine and Pharmacy, 700115 Iasi, Romania; bogdan.dobrovat@yahoo.com (B.I.D.); danisiahaba@yahoo.com (D.H.); 5Department of Implantology, Removable Prostheses, Dental Prostheses Technology, Faculty of Dental Medicine, “Grigore T. Popa” University of Medicine and Pharmacy, 700115 Iasi, Romania; cosmin.bida@umfiasi.ro; 6Department of Intensive Care Unit, “Saint Mary” Emergency Children Hospital, 700309, Faculty of Medical Bioengineering, “Grigore T. Popa” University of Medicine and Pharmacy, 700115 Iasi, Romania; valentin_munteanu2022@yahoo.com

**Keywords:** dental appliance, orthodontics, obstructive sleep apnea, pediatric

## Abstract

*Background and objectives*: Obstructive sleep apnea (OSA) in children is a debilitating disease, difficult to treat. Dental appliances have been proposed as a valid therapy for improving functional outcomes with good compliance rates. Herein, we aimed to perform a meta-analysis comparing clinical outcomes between OSA children treated with dental appliances versus controls. *Materials Methods*: The study was registered with PROSPERO. A systematic search was performed for all comparative studies examining outcomes in pediatric patients who underwent treatment of OSA with oral appliances versus controls. Data was extracted and analyzed using a random effects model via Rev Man 5.3. *Results*: Six studies including 180 patients were analyzed split into two groups: patients treated with dental appliances (n = 123) and the controls (n = 119). Therapy with dental appliances was shown to significantly improve the apnea–hypopnea index (*p* = 0.009) and enlarge the superior posterior airway space (*p* = 0.02). Maxilla-to-mandible measurements were not significantly different between the two groups, nor was the mean SO_2_ (*p* = 0.80). *Conclusions*: This is the most updated meta-analysis assessing the role of dental appliances for OSA in children; it shows that such devices can improve functional outcomes by decreasing the apnea–hypopnea index.

## 1. Introduction

Obstructive sleep apnea (OSA) is defined as an abnormal, recurrent collapse of the upper airway resulting in disrupted ventilation and affecting normal sleep patterns [1,2]. Children affected by OSA usually present with habitual snoring, restless sleep, behavioral changes, morning headaches, and interrupted sleep patterns, and may be misdiagnosed with behavioral diseases before the possibility of OSA is raised [3]. OSA is a debilitating disease proven to independently increase all-cause mortality by 1.9 times [4]. While obesity is the main modifiable risk factor for the development of obstructive sleep apnea (OSA) in adults, in children OSA is associated with upper airway variability and can equally affect both boys and girls, while in adults OSA is more frequent in males [5]. There is, however, a certain variability of severity among children with OSA, often not related to the apparent risk factors. Low-grade systemic inflammation has been found in certain populations by increased levels of pro-inflammatory cytokines such as IL-6, TNF-α, and IFN-γ. It is also amplified by the presence of obesity, which in itself causes chronic inflammation. Plasma levels of monocyte chemoattractant protein 1 and plasminogen activator-inhibitor 1 have been found to be increased in obese individuals with and without sleep apnea and are also linked to metabolic syndrome and atherosclerosis. The latter conditions are strongly linked to vascular endothelial dysfunction, which is greater in children with OSA. Low-grade systemic inflammation causes significant neurocognitive deficit, and it seems to be inversely correlated with levels of endothelial progenitor cells, with stromal cell-derived factor-1, and directly with level of DNA methylation involving the FOXP3 gene, which is responsible for the activity of T regulatory cells. Genetic polymorphism seems also to play a role. Reduced NADPH oxidase activity seems to be associated with milder cognitive deficit, with opposite effects for apolipoprotein E alleles [6].

Thus, management of OSA in pediatric populations deserves special consideration, as it is not associated with risk factors which can be modified through lifestyle changes.

Mild forms of pediatric OSA can be managed with a combination of intranasal glucocorticoids and leukotriene inhibitors, which have been shown to improve the apnea–hypopnea index (AHI) [7,8]. In children with tonsillar hypertrophy, adenotonsillectomy relieves symptoms in the majority of patients [9] and is a reliable option if one accepts the associated surgical morbidity and risk of recurrence. Although an effective method in adults, in children continuous positive airway pressure (CPAP) is difficult to implement due to poor compliance [10,11]. Orthodontic correction of class II deformities has been shown to be a valuable option and can be performed either through rapid palatal expansion or through oral devices with promote mandibular advancement. To correct palatal contraction, which has been shown to increase the risk of OSA development, rapid palatal expansion can be an effective measure through widening of the oropharyngeal space, resulting in an increased total nasal volume [12].

Several dental/oral appliances have been designed and tested with seemingly good results in improving oxygen delivery by reducing episodes of airway collapse [13,14,15,16,17,18,19,20]. These appliances are constructed to correct upper airway anomalies and to maintain airway patency by stabilizing the soft palate and by increasing the longitudinal diameter of the posterior oropharyngeal airway through mandibular protrusion. Herein we aim to perform a systematic review and meta-analysis analyzing outcomes of such devices in the management of pediatric OSA.

## 2. Materials and Methods

### 2.1. Literature Search and Study Selection

The study protocol was registered with PROSPERO (International Prospective Register of Systematic Reviews). The study ID is CRD42023420703. A systematic search of the PubMed and EMBASE databases was performed for all studies examining clinical outcomes in pediatric patients who underwent treatment of OSA with oral/dental appliances. The following search algorithm was used: (oral OR dental) AND (appliance OR apparatus OR device) AND (obstructive sleep apnea) AND (children OR pediatric). The preferred reporting items for systematic reviews and meta-analyses (PRISMA) guidelines were used as a search protocol, and the PRISMA checklist was followed to perform the methodology [21] (Figure 1).

Inclusion criteria were used according to the problem, intervention, comparison, and outcome (PICO) formula. Only studies that used custom-made oral correction devices were used. Treatment of OSA through adenotonsillar surgery was not reviewed herein. Both congenital and acquired class II malocclusions and mandibular retrognathism were included. Patients without skeletal deformities but with confirmed OSA were also included to assess functional improvement. The latest search was performed on 9 May 2023. Two of the authors (DM and SM) assessed the titles and abstracts of studies found in the search, and the full texts of potentially eligible trials were reviewed. Disagreements were resolved by consensus-based discussion. The Newcastle–Ottawa scale (Table 1) and the ROBINS-I tool [22] (Figure 2) were used to quantify the quality of eligible studies. The references of full texts were further screened for additional eligible studies. The corresponding author was contacted to clarify data extraction if additional information was necessary.

### 2.2. Eligibility Criteria

Our survey included studies written in English which included comparative data between children with OSA managed with oral appliances versus controls. Studies which compared baseline measurements versus posttreatment measurements in the same cohort were also considered. The primary endpoints were cephalometric measurements including Sella-Nasion-A (SNA), Sella-Nasion-B (SNB), A-Nasion-B (ANB), superior-posterior airway space (SPAS), middle airway space (MAS), and inferior airway space (IAS). Also, improvement in OSA symptoms was analyzed by comparing the apnea–hypopnea index (AHI) and the mean SO_2_ values. Studies without comparative data were not included. 

### 2.3. Data Extraction and Outcomes

For each eligible study, the following data were recorded: authors’ names, journal, year of publication, study type, total number of patients, number of patients included in each group (study versus control), mean age of the included patients, gender, OSA stage, type of appliance used, cephalometric measurements, polysomnography measurements, patient-reported outcomes, and duration of follow-up. All the data were collected in an Excel database, which was then transferred to Review Manager 5.3 for further analysis. The two groups were compared in a meta-analytical model considering three blocks of variables: (i) group comparability (mean age, mean BMI); (ii) cephalometric measurements (SNA, SNB, ANB, SPAS, MAS, IAS); and (iii) polysomnographic measurements (AHI, mean SO_2_).

### 2.4. Statistical Analysis

As previously demonstrated [23,24,25], random-effects models were used to measure all the pooled outcomes as described by Der Simonian and Laird [26]. For dichotomous variables, the odds ratio (OR) was estimated with its variance and 95% confidence interval (CI), while for continuous data, the mean difference was used at 95% CI. The random effects analysis weighted the natural logarithm of each study’s OR by the inverse of its variance plus an estimate of the between-study variance in the presence of between-study heterogeneity. Heterogeneity between ORs for the same outcome between different studies was assessed using the I^2^ inconsistency test and the chi-square-based Cochran’s Q statistic test, in which *p* < 0.05 is taken to indicate the presence of significant heterogeneity. The analyses were conducted using Review Manager 5.3.

## 3. Results

### 3.1. Eligible Studies

Six studies [15,16,17,18,19,20] containing data on cephalometric and polysomnographic outcomes after dental appliance application in OSA patients were included (Table 1). The initial search found 619 studies. After excluding duplicates and unrelated studies based on abstract triage, 18 full texts were assessed for eligibility, out of which 6 matched the inclusion criteria and were analyzed. The year of publication of the included studies ranged from 2004 to 2022. There was one randomized controlled trial (RCT); three case control studies; and two prospective, observational studies. The total number of included patients was 180, split into two groups: the study group (SG), in which dental appliances were used; and the control group (CG). 

### 3.2. Overview of Studies

All the studies used a custom-made mandibular advancement monobloc made of either splint or resin. The devices were designed by orthodontic technicians so as to correct mandibular malposition and were made of two plates, one each for the upper and lower teeth. The designed plates matched each patient’s bite, but the lower piece was activated to ensure the forward protrusion of the mandible. The mandibles were advanced forward to achieve vertical alignment of the upper and lower teeth. In most cases, the devices were worn full-time except at mealtimes. The two groups were compared through cephalometric measurements or by polysomnography to assess the AHI and mean SO_2_ values. The follow-up period expanded from 3 weeks to 18.3 months. Table 2 provides an overview of the main comparisons for each study.

### 3.3. Group Comparability

Mean age and BMI are depicted in Table 1. Three studies [15,17,19] describing 102 patients reported mean age and BMI in their respective cohorts. No significant difference in age was found between the two groups (mean difference: 0.52 years, 95% CI: [0.41, 1.46], *p* = 0.27, Chi^2^ = 6.34, I^2^ = 68%). There was no significant difference in BMI between the two groups despite a mean difference of 2.55 in favor of the SG, 95% CI: [0.32, 5.43], *p* = 0.08, Chi^2^ = 14.02, I^2^ = 86%) (Figure 3A,B).

### 3.4. Cephalometric Measurements

#### Maxilla to Mandible Measurements

Four studies [15,17,18,20] describing 203 patients provided data on maxilla-to-mandible lengths by measuring the SNA, SNB, and ANB in the two groups of patients. No significant difference in SNA was found between the two groups (mean difference: 0.13, 95% CI: [0.75, 1.00], *p* = 0.78, Chi^2^ = 1.51, I^2^ = 0%) (Figure 4A). No significant difference in SNB was found between the two groups despite the CG having a wider SNB by 1.35, 95% CI: [0.17, 2.87], *p* = 0.08, Chi^2^ = 8.98, I^2^ = 67%) (Figure 4B). No significant difference in ANB was found between the two groups (mean difference: 0.02, 95% CI: [3.06, 3.09], *p* = 0.99, Chi^2^ = 46.77, I^2^ = 96%) (Figure 4C).

### 3.5. Upper Airway Measurements

Three studies [17,18,20] describing 173 patients provided data on upper airway measurements by calculating the SPAS, MAS, and IAS in the two groups of patients. The SG was associated with significantly longer SPAS (mean difference: 0.26 cm, 95% CI: [0.03, 0.48], *p* = 0.02, Chi^2^ = 15.04, I^2^ = 87%) (Figure 5A). No significant difference in MAS length was found between the two groups (mean difference: 0.31, 95% CI: [0.23, 0.85], *p* = 0.26, Chi^2^ = 15.65, I^2^ = 94%) (Figure 5B). No significant difference in IAS length was found between the two groups (mean difference: 0.00, 95% CI: [0.12, 0.12], *p* = 0.94, Chi^2^ = 0.01, I^2^ = 0%) (Figure 5C).

### 3.6. Polysomnographic Measurements

Four studies [15,16,19,20] describing 180 patients provided data on polysomnographic measurements in the two groups of patients. The AHI and mean SO_2_ were compared. The AHI was significantly lower in the SG with a mean difference of 5.44, 95% CI: [1.35, 9.54], *p* = 0.009, Chi^2^ = 63.50, I^2^ = 95%) (Figure 6A). No significant difference in mean SO_2_ was found between the two groups (mean difference: 0.09, 95% CI: [0.57, 0.74], *p* = 0.80, Chi^2^ = 8.65, I^2^ = 77%) (Figure 6B). 

## 4. Discussion

This meta-analysis demonstrates that using dental appliances for OSA in children improves symptomatic outcomes by reducing the AHI and increasing the SPAS, without significantly changing the cephalometric measurements or significantly increasing the mean SO_2_. 

OSA should be diagnosed at an early stage due to its systemic complications such as learning and growth impairment, behavioral changes, and cardiovascular involvement [3]. The main risk factors are adenotonsillar hypertrophy, allergic rhinitis, obesity, and the associated chronic inflammation, as well as craniofacial anomalies, neuromuscular disorders, and multiple pregnancy [27]. OSA can be often difficult to diagnose in the pediatric population due to a lack of specific symptoms. Although snoring is the most common symptom, hyperactivity or inattention are more specific. It seems that learning difficulties, lower executive function, poor memory, or hyperactivity disorders are often signs of underlying OSA. Some nocturnal symptoms, apart from snoring, can be often noted by parents, such as gasping, noisy or restless sleeping, mouth breathing, apneas, enuresis, or parasomnia. In some cases, the condition may be overlooked until the child fails to thrive or develop cardiovascular morbidity such as secondary or pulmonary hypertension or cor pulmonale with signs of right-heart failure. All these symptoms develop as a consequence of intermittent hypoxia, with subsequent increasingly labored breathing, abnormal growth hormone secretion patterns, and anomalies in the development of the prefrontal cortex [28].

Prior to initiation of treatment, a thorough clinical evaluation of the pediatric OSA patient is required. In healthy children with OSA, cephalometric studies have shown that they have a narrower posterior airway space, with anomalies of mandible occlusal and vertical orientation. In cases of adenoid hypertrophy, the child tends to extend his or her head to increase the space of the posterior airway, which pulls toward the mandible, resulting in mouth breathing and transverse maxillary constriction. In children with a genetic syndrome, the associated craniofacial changes are more prominent, hence OSA is more frequent and potentially of increased severity. Around 180 genetic syndromes seem to be associated with craniosynostosis, out of which 68% may be diagnosed with OSA [29]. Premature fusion of cranial sutures is most often associated with Apert syndrome (FGFR2), Muenke syndrome (FGFR3), Crouzon syndrome (FGFR2), Pfeiffer syndrome (FGFR2, FGFR1), and Saethre-Chotzen syndrome (TWIST1) [30]. The anomalies associated with the anterior skull base impact the posterior skull base angulation, leading to a steep mandible inclination causing a narrow posterior airspace. A meta-analysis showed a prevalence of OSA of up to 76% in children with Down syndrome, as they are associated with multiple predisposing factors such as midfacial hypoplasia, macroglossia, and poor muscle tone [31]. A high prevalence was also observed in Ehlers-Danlos syndrome. Forty-two per cent of the pediatric patients were diagnosed with OSA, which is a higher percentage than in children with oral anomalies such as cleft palate, but lower than for Pierre Robin or Down syndrome. This high prevalence is explained by flaccid tissue and cartilages in the pharyngeal anatomy, increasing collapsibility [32]. Hence, multiple diagnostic tools can be used, such as pediatric sleep questionnaires, facial imagistic investigations, nocturnal oximetry, or ambulatory polysomnography. A multidisciplinary approach by ENTs, orthodontists, and pediatric clinicians should be employed, as management is still challenging [3].

Adenotonsillectomy is a potentially curative solution, with a high rate (83%) of resolution of polysomnographic changes in children without other associated comorbidities. It requires, however, long-term follow-up, as some studies have reported up to a 47% recurrence of symptoms [33]. Positive airway pressure (PAP) therapy is another alternative treatment which has been shown to be superior to dental appliances in improving functional outcomes. Indication is generally established following evaluation through a sleep study, ideally in a specialized sleep center. Either continuous (CPAP) or bilevel (BIPAP) pressures can be used, with a higher preference for CPAP. Although it has the role of a pneumatic split preventing the soft airway tissue from collapsing, patients are less compliant due to local discomfort caused by nasal congestion, epistaxis, eye irritation, or skin abrasion [14]. Hence, some patients prefer dental procedures. 

To our knowledge, this is the first meta-analysis to compare cephalometric and polysomnographic measurements in a pediatric cohort in which dental appliances were used for OSA treatment. Many children with OSA also have craniofacial deformities. This has been especially reported in patients with genetic syndromes, such as Down, Prader-Wili, or Beckwith–Wiedemann, as well as achondroplasia and Noonan, Ehlers–Danlos, or Ellis-van Creveld syndromes. Retrognathia, reduced antero-posterior length of the bony pharynx, reduced cranial base angle, soft tissue enlargement, and abnormal muscular tone are some of the anatomical abnormalities impeding normal breathing patterns; oral/dental devices could be used here as a measure to relieve functional disturbances [34].

There are two main types of orthodontic interventions that can be performed [35,36,37,38]. A mandibular repositioning device increases the area of the hypopharynx by moving anteriorly the mandible and the base of the tongue. This can be used in patients with mild to moderate sleep apnea or in cases of poor tolerance of PAP in severe forms. One study including 19 subjects showed a 68.4% rate of successful treatment [34], similar to other investigators [39,40]. This is probably due to higher compliance, but treatment duration should be at least six months. Interestingly, in patients with longer treatment duration, a correction in facial anomalies can be achieved especially if the children are at their peak growth. This device can be considered safe, effective, and low-cost, especially in patients with mild forms of OSA, but polysomnography surveillance is indicated in severe forms of the disease [13]. Rapid maxillary expansion can be used to obtain correction of posterior crossbites by widening the maxilla leads, for improved coordination of the dental arches, for decreased nasal resistance to air flow, and for better tongue position. This is achieved by securing a dental device with an expansion screw over the maxillary teeth. Subsequently, an increase in the efficiency of each respiration, a widening of the oropharyngeal space, and even tonsillar downsizing should be observed. This method has been used in children with dental malocclusion and maxillary restriction showing resolution of abnormal polysomnography parameters [41]. Other devices can improve airway patency by pushing the tongue and mandible anteriorly and show good results with regard to OSA symptoms and parameters [19]. Our study agrees that dental appliances should be able to improve clinical outcomes and quality of sleep by reducing AHI, with some studies supporting these findings even long-term, up to 14 years [42]. In addition to AHI, these devices appear to improve patients’ quality of life by correcting sleep patterns and enhance cognitive function by reducing daytime somnolence, or irritability [43,44].

The main limitation of this meta-analysis is the lack of valuable data and the small number of RCTs. Only six studies provided enough data for systematic analysis, and even in these six studies, important variables were missing; these included cephalometric measurements to confirm skeletal correction in some studies and functional assessment through PSG in others. However, because OSA has a low incidence in the general pediatric population and because correction devices are not used in a consistent manner, it is unlikely that well-powered RCTs will be published in the near future. Despite being limited by heterogenous data and a small number of included patients, this study is the most up-to-date and complete case-control comparison in patients with OSA treated with oral appliances.

## 5. Conclusions

The use of dental appliances in children for the management of obstructive sleep apnea is shown to be effective in improving breathing patterns by reducing the apnea–hypopnea index. Predominantly used in patients with associated craniofacial abnormalities or in mild to moderate forms, these devices are suitable as a first-line management or as an alternative to an invasive treatment. More studies are needed to compensate for the small cohort and heterogenous data available in the current literature.

## Figures and Tables

**Figure 1 medicina-59-01447-f001:**
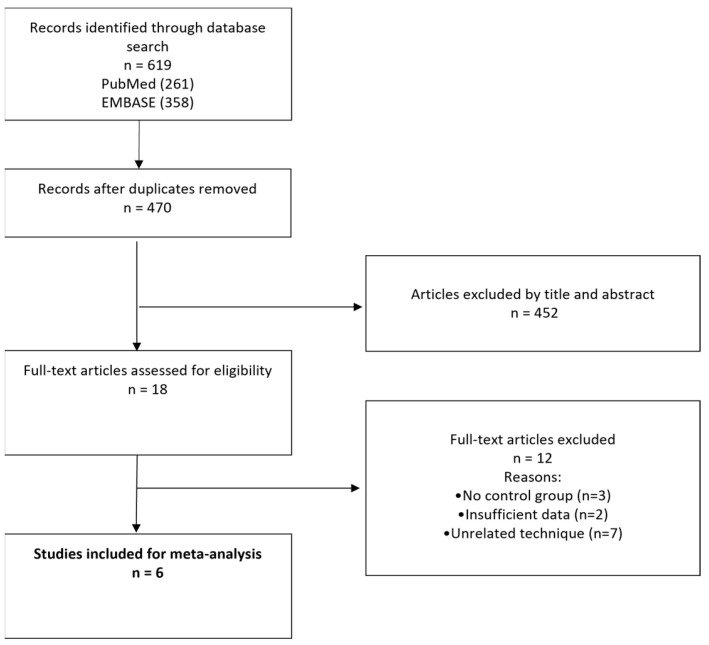
PRISMA flowchart for study selection and final inclusion.

**Figure 2 medicina-59-01447-f002:**
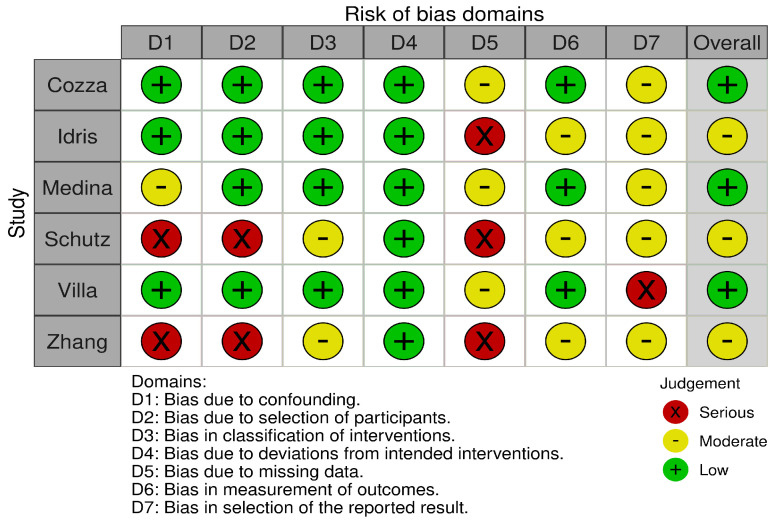
ROBINS-I risk of bias assessment. Assessment of risk of bias was performed by two authors (DM and SM). Each study was classified as low/moderate/serious/critical risk for each of the seven domains. Disagreements were resolved via consensus.

**Figure 3 medicina-59-01447-f003:**
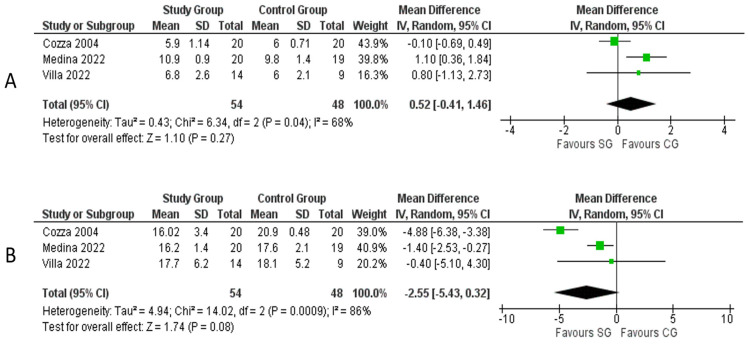
Meta-analysis of group comparability: (**A**) age; (**B**) BMI. **Legend**: Each study is shown by the point estimate of the mean difference and 95% confidence interval (CI); the combined mean difference and 95% CIs by random effects calculations are shown by diamonds. (**A**) SG versus CG and mean age (n = 102, *p* = 0.27; test for heterogeneity Cochran Q: 6.34, df: 2, *p* = 0.04, I^2^: 68%). (**B**) SG versus CG and mean BMI (n = 102, *p* = 0.08; test for heterogeneity Cochran Q: 14.02, df: 2, *p* = 0.0009, I^2^: 86%) [15,17,19].

**Figure 4 medicina-59-01447-f004:**
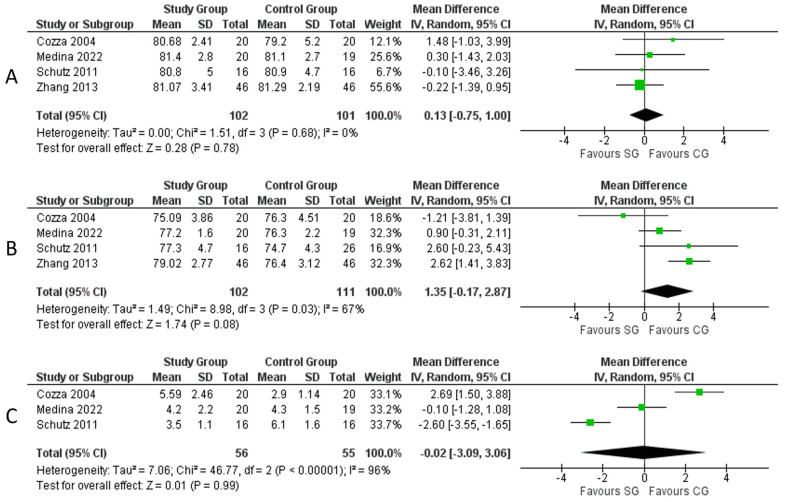
Meta-analysis of cephalometric measurements: (**A**) SNA; (**B**) SNB; (**C**) ANB. **Legend**: Each study is shown by the point estimate of the mean difference and the 95% confidence interval (CI); the combined mean difference and 95% CIs by random effects calculations are shown by diamonds. (**A**) SG versus CG and SNA (n = 203, *p* = 0.78; test for heterogeneity Cochran Q: 1.51, df: 3, *p* = 0.68, I^2^: 0%). (**B**) SG versus CG and SNB (n = 203, *p* = 0.08; test for heterogeneity Cochran Q: 8.98, df: 3, *p* = 0.03, I^2^: 67%). (**C**) SG versus CG ANB (n = 111, *p* = 0.99; test for heterogeneity Cochran Q: 46.77, df: 2, *p* < 0.00001, I^2^: 96%) [15,17,18,20].

**Figure 5 medicina-59-01447-f005:**
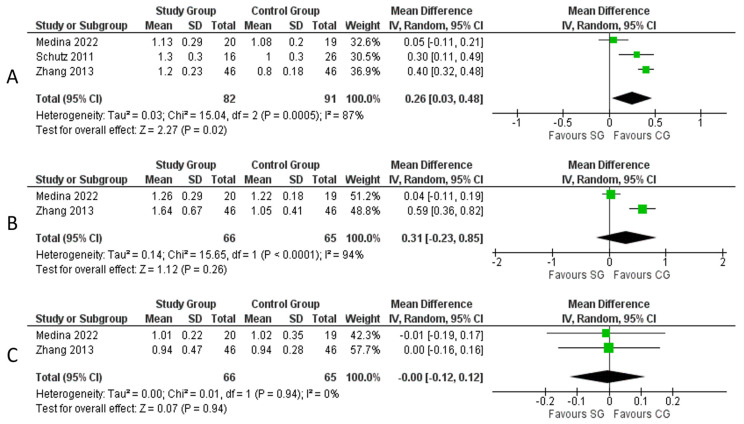
Meta-analysis of upper airway measurements: (**A**) SPAS; (**B**) MAS; (**C**) IAS. **Legend**: Each study is shown by the point estimate of the mean difference and 95% confidence interval (CI); the combined mean difference and 95% CIs by random effects calculations are shown by diamonds. (**A**) SG versus CG and SPAS (n = 173, *p* = 0.02; test for heterogeneity Cochran Q: 15.04, df: 2, *p* = 0.0005, I^2^: 87%). (**B**) SG versus CG and MAS (n = 131, *p* = 0.26; test for heterogeneity Cochran Q: 15.65, df: 1, *p* < 0.0001, I^2^: 94%). (**C**) SG versus CG IAS (n = 131, *p* = 0.94; test for heterogeneity Cochran Q: 0.01, df: 1, *p* = 0.94, I^2^: 0%) [17,18,20].

**Figure 6 medicina-59-01447-f006:**
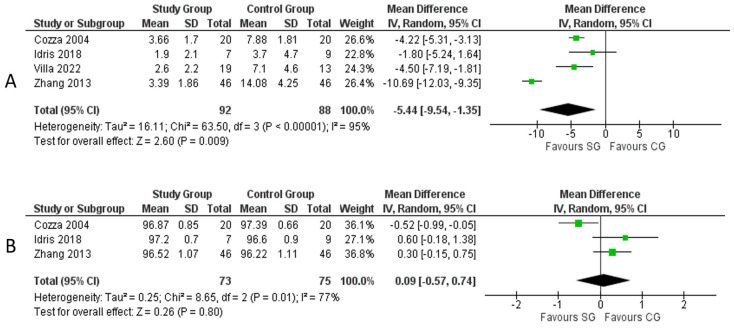
Meta-analysis of polysomnographic measurements: (**A**) AHI; (**B**) mean SO_2_. Legend: Each study is shown by the point estimate of the mean difference and 95% confidence interval (CI); the combined mean difference and 95% CIs by random effects calculations are shown by diamonds. (**A**) SG versus CG and AHI (n = 180, *p* = 0.009; test for heterogeneity Cochran Q: 63.50, df: 3, *p* < 0.00001, I^2^: 95%). (**B**) SG versus CG and mean SO_2_ (n = 148, *p* = 0.80; test for heterogeneity Cochran Q: 8.65, df: 2, *p* = 0.01, I^2^: 77%) [15,16,19,20].

**Table 1 medicina-59-01447-t001:** Eligible studies and their characteristics. Key: SG, study group; CG, control group; BMI, body mass index; NOS, Newcastle–Ottawa Scale; RCT, randomized control trial.

Author	Year	Type of Study	Total No. of Patients		Age (Mean)			BMI (Mean)			NOS
			SG	CG							
Cozza [15]	2004	case control	40		SG	CG		SG		CG	8
			20	20	5.9	6		16.02		20.9	
Idris [16]	2018	RCT	16		9.8			20.4			7
			7	9							
Medina [17]	2022	case control	39		SG	CG		SG	CG		8
			20	19	10.9	9.8		16.2	17.6		
Schutz [18]	2011	prospective	16		12.6			18.3			6
			16	16							
Villa [19]	2022	RCT	23		SG		CG	SG	CG		8
			14	9	6.8		6.0	17.7	18.1		
Zhang [20]	2013	prospective	46		9.7			18.1			6
			46	46							

**Table 2 medicina-59-01447-t002:** Overview of the included studies. Key: AHI, apnea–hypopnea index; PSG, polysomnography.

Author	Device Used	Time of Wear	Follow-Up	Test Used	AHI Improvement		SO_2_ (%) Improvement	
					Device Not Used	Device Used	Device Not Used	Device Used
Cozza [15]	Mandibular monobloc	Nights only	6 months	PSG	7.88	3.66	97.39	96.87
Idris [16]	Mandibular twin block	Full time	3 weeks	PSG	3.7	1.9	96.6	97.2
Medina [17]	acrylic-splint Andresen mandibular activator	Full time	18.3 months	Cephalometric analysis only				
Schutz [18]	acrylic-splint Herbst mandibular activator	Full time	12 months	Cephalometric analysis only				
Villa [19]	acrylic resin mandibular monobloc	Full time	6 months	PSG	7.1	2.6	NR	
Zhang [20]	acrylic resin twin block	Full time	10.8 months	PSG	14.08	3.39	96.22	96.52

## Data Availability

Further data available by contacting the corresponding author: morarasu.bianca.codrina@gmail.com/emma.marciuc@gmail.com.

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
