# Peer review of "Dental Appliances for the Treatment of Obstructive Sleep Apnea in Children: A Systematic Review and Meta-Analysis"

_medicina, 2023, doi:10.3390/medicina59081447_

Round 1

Reviewer 1 Report

Comment to the authors_ Medicina

Thank you for inviting me to review the article entitled “Dental appliances for the treatment of obstructive sleep apnea in children: a systematic review and meta-analysis”. This is a systematic review and meta-analysis on the effect of dental appliance in the treatment of OSA in children. 

The study and the analysis are well conducted, very well organized. However, it lacks in the following items here identified. 

Major comments: 

- The authors should summarize the results also in a table, where they should review the studies to include other variables, like time of follow-up of the intervention, where they show which treatment was devoted the control group, baseline AHI, baselins SpO2, which instrumental exam the two groups were tested with (PSG, etc), which appliance was utilized, and all the other parameters that the authors might think important. This in addition to the tables presenting the meta-analysis data (indeed, the title of the paper is systematic review, beside the meta-analysis). They should also include which papers have been qualitatively compared in a systematic review approach. 

- The authors should also devote a paragraph in the results to report which dental appliances have been used. Are these mandibular advancement devices? Or are these palatal expanders? Which appliances were utilized? 

- in the exclusion criteria, the authors should mention if they accepted paper including adenotonsillar surgery or not. Did the authors also include syndromic patients or were they excluded? This needs to be specified

- I suggest that the authors (either in the discussion in lines 202-206 or in the introduction) mention which the symptoms and signs are of children presenting with OSA

- In the limitations, the authors need to include that these findings are based on small sample of included studies, considering that not all the six include data on all the variables examined. 

Minor comments: 

- page 2, line 47: refrain from contraction in scientific manuscript (“aren’t”) 

- pag 2, line 57: “These appliances are contructed to correct uppe airway deformities”. I would replace the word “deformity” with a more appropriate term. It is only constricted, not deformed. 

- the authors should also allude to the rapid palatal expander in their introduction. 

- Table 1: it is not very clear the second row under the total number of patients. I suggest that the authors introduce a second row (under No. of patients) with SG vs CG. Also, the authors need to reference the studies under the authors (put the number of reference in the Reference list)

Reviewer 2 Report

The aim of the present SR and MA is to analyze the outcomes of the devices used in the management of pediatric OSA.

The usefulness of the existing devices remains controversial. It really addresses a gap in this field.

It is a very well-designed SR and MA. Although, the authors should provide more details about the devices and design, they could probably add some figures of them, if available.

In line 250 at the end of discussion the authors state the main limitations of the MA, which are the small number of included patients and the heterogeneity. Conclusions are presented adequately.

They should consider to include the following references concerning sleep apnea.

Tsolakis, I.A.; Palomo, J.M.; Matthaios, S.; Tsolakis, A.I. Dental and Skeletal Side Effects of Oral Appliances Used for the Treatment of Obstructive Sleep Apnea and Snoring in Adult Patients—A Systematic Review and Meta-Analysis. J. Pers. Med. 2022, 12, 483.

Tsolakis IA, Venkat D, Hans MG, Alonso A, Palomo JM. When static meets dynamic: Comparing cone-beam computed tomography and acoustic reflection for upper airway analysis. Am J Orthod Dentofacial Orthop. 2016 Oct;150(4):643-650.

Tsolakis, I.A.; Kolokitha, O.-E. Comparing Airway Analysis in Two-Time Points after Rapid Palatal Expansion: A CBCT Study. J. Clin. Med. 2023, 12, 4686

Tsolakis, I.A.; Kolokitha, O.-E.; Papadopoulou, E.; Tsolakis, A.I.; Kilipiris, E.G.; Palomo, J.M. Artificial Intelligence as an Aid in CBCT Airway Analysis: A Systematic Review.
Life 2022, 12, 1894

Round 2

Reviewer 2 Report

The authors made all the appropriate changes. It is a very well structured SR and MA.